# Perceptions about Telemedicine among Populations with Chronic Diseases amid COVID-19: Data from a Cross-Sectional Survey

**DOI:** 10.3390/ijerph19074250

**Published:** 2022-04-02

**Authors:** Miah Md. Akiful Haque, Yasmin Jahan, Zara Khair, Michiko Moriyama, Md. Moshiur Rahman, Mohammad Habibur Rahman Sarker, Shamsun Nahar Shaima, Sajeda Chowdhury, Kazi Farhana Matin, Ishrat Jahan Karim, Mostafa Taufiq Ahmed, Syed Zakir Hossain, Md. Adnan Hasan Masud, Mohammad Golam Nabi, Asma Binte Aziz, Mohiuddin Sharif, Md. Forhadul Islam Chowdhury, Kaniz Laila Shams, Nusrat Benta Nizam, Taiyaba Tabassum Ananta, Md. Robed Amin, Mohammad Delwer Hossain Hawlader

**Affiliations:** 1Department of Public Health, North South University, Dhaka 1229, Bangladesh; miah.haque@northsouth.edu (M.M.A.H.); kazi.farhanamatin@northsouth.edu (K.F.M.); mohammad.hawlader@northsouth.edu (M.D.H.H.); 2Public Health Professional Development Society (PPDS), Dhaka 1215, Bangladesh; 3Graduate School of Biomedical and Health Sciences, Hiroshima University, Hiroshima 734-8553, Japan; dr.yasminjahan@gmail.com (Y.J.); zara.oikee@gmail.com (Z.K.); morimich@hiroshima-u.ac.jp (M.M.); habibur.rahman@icddrb.org (M.H.R.S.); sajeda.chowdhury@gmail.com (S.C.); 4International Centre for Diarrhoeal Disease Research, Bangladesh (icddr,b), Dhaka 1212, Bangladesh; shamsun.shaima@icddrb.org; 5Sylhet MAG Osmani Medical College, Sylhet 3100, Bangladesh; drijkog@gmail.com (I.J.K.); drtaufiqns@gmail.com (M.T.A.); 6Dhaka Medical College, Dhaka 1000, Bangladesh; bibosi1@yahoo.com (S.Z.H.); mohiuddinsharif.fmc@gmail.com (M.S.); robedamin@yahoo.com (M.R.A.); 7Haematology Department, Bangabandhu Sheikh Mujib Medical University (BSMMU), Dhaka 1000, Bangladesh; dr.adnan.hasan@gmail.com; 8Right Turn, Kalapara 8650, Bangladesh; gnabi1969@gmail.com; 9International Vaccine Institute, Seoul 08826, Korea; tulykhan44@gmail.com (A.B.A.); nforhad.b24@gmail.com (M.F.I.C.); 10Bangladesh Institute of Health Sciences (BIHS) General Hospital, Dhaka 1207, Bangladesh; kanizlaila75@gmail.com (K.L.S.); fnila.b25@gmail.com (N.B.N.); 11Department of Food and Nutrition, College of Home Economics, Dhaka 1205, Bangladesh; ananta.tabassum@gmail.com

**Keywords:** perception, telemedicine, chronic disease, COVID-19, Bangladesh

## Abstract

Chronic diseases, including non-communicable diseases (NCDs), have arisen as a severe threat to health and socio-economic growth. Telemedicine can provide both the highest level of patient satisfaction and the lowest risk of infection during a pandemic. The factors associated with its usage and patient adherence are not visible in Bangladesh’s resource-constrained settings. Therefore, this study aimed to identify perceptions about telemedicine among populations with chronic diseases amid the COVID-19 pandemic. A closed-ended self-reported questionnaire was created, and the questionnaire was written, reviewed, and finalized by a public health investigator, a psychiatrist, and an epidemiologist. The data for this study were collected from individuals using simple random sampling and snowball sampling techniques. Ethics approval was granted, and written/verbal consent was taken before interviews. Most of the participants showed a positive attitude towards telemedicine. People aged 35–54 years old and a higher level of education were less frequently associated with willingness to receive telemedicine services for current chronic disease (WRTCCD) than their counterparts. People living in urban areas and lower-income participants were more strongly associated with WRTCCD. Additionally, people who did not lose their earnings due to the pandemic were less strongly associated with WRTCCD. However, the main strength of this research is that it is a broad exploration of patient interest in several general forms of telehealth. In Bangladesh, there are many opportunities for telemedicine to be integrated into the existing healthcare system, if appropriate training and education are provided for healthcare professionals.

## 1. Introduction

Chronic diseases include non-communicable diseases (NCDs) such as hypertension, diabetes, cancer, cardiovascular disease (e.g., heart attack, stroke), and chronic arthritis [1]. NCDs have arisen as a severe threat to health and socio-economic growth [2]. NCDs have significantly increased premature deaths and have placed a twofold burden of disease on the health system (previously existing communicable diseases, plus newly added non-communicable diseases) via increased service demand and overall treatment expense [2,3]. Moreover, poor patient treatment adherence increases both the mortality and morbidity burden, and the healthcare utilization and cost of NCDs. The impact of poor adherence has grown in tandem with the rise in prevalence of NCDs, and this issue is anticipated to be worsened further during the COVID-19 pandemic [4]. Further, the pandemic has limited consultation availability and time, and led to a reduction in in-person examinations [3].

Telemedicine can be a good solution when distance impairs the delivery of proper care to a patient. The aims of telemedicine are to allow the long-term management of chronic illnesses, and to provide immediate information to give an early warning of possible diseases—both of which meet the goal of preventive healthcare [5]. During the pandemic, the care of people with severe illness and their families required impeccable social distancing for their protection, and for the protection of healthcare professionals, who were so critically needed [6]. Therefore, telemedicine is a proven modality for delivering palliative care to the most vulnerable people.

During the COVID-19 pandemic, access to telemedicine has been increased for patients, along with other improvements to patient access, to help ensure continuity of care across a variety of different patient groups, including NCDs [7]. The COVID-19 pandemic spurred the medical profession to embrace telemedicine, and protocols were swiftly altered to ensure patient care continued during the pandemic [8].

Standard of care refers to the consistent and connected care that a patient receives across time, in accordance with their health needs and personal circumstances. It is recognized as a fundamental value of patient healthcare and serves as a proxy for the level of care provided in general practice. Additionally, benefits are noted from the patients’ standpoint. Patients favor continuity of care [9], particularly those with chronic diseases or high healthcare utilization [10]. As such, telemedicine can provide the highest level of patient satisfaction and, at the same time, the lowest risk of infection in the midst of a pandemic, by continuing the patient’s care under the same specialist remotely. As the quick transmission of COVID has increased fear among individuals amid the pandemic, and it has become more hazardous for a wide range of patients (including those with COVID-19) to go to a specialist or medical clinic to receive therapy, telemedicine administrations have emerged as a solution.

Bangladesh’s internet penetration rate was 73.06% in November 2021 [11]. This coverage enables the use of web network services (such as telemedicine, video calls, and emails) to connect with individuals and provide more tailored healthcare, which may improve adherence and illness outcomes, hence reducing the burden on the health system and healthcare costs [12]. However, telehealth service consumption has not yet achieved its full potential. Additionally, patients’ desire to employ such an intervention as an extra treatment option, and the factors associated with its usage and patient adherence, are not yet visible in Bangladesh’s resource-constrained settings. Patients suffering from chronic diseases may place a high value on routine check-ups or hospital visits; however, the COVID-19 pandemic has affected them by reducing access to care. There is considerable evidence that telemedicine, if offered as a part of routine follow-up consultations, can help better manage patients with chronic NCDs, democratizing the continuity of care to prevent an increase in chronic disease-related morbidity and mortality [13,14]. Additionally, knowing what interests patients in participating in various forms of telehealth is a crucial first step if telehealth is to play an important role in effectively assisting patients with NCDs. Therefore, this study aimed to identify the perceptions about telemedicine among populations with chronic diseases amid COVID-19.

## 2. Materials and Methods

### 2.1. Study Design

We conducted a descriptive cross-sectional survey between September and November 2020. The majority of the data for this study were collected through face-to-face interviews at public tertiary hospitals in Bangladesh using convenient sampling. A small proportion of study participants, on the other hand, were interviewed over the phone using snowball sampling procedures.

### 2.2. Participants and Setting

The following criteria were used to determine eligibility for participation in the study: (i) currently resides in Bangladesh (aged 18 years and above) of either gender; (ii) diagnosed with chronic diseases such as diabetes, chronic kidney disease, malignancy, or cardiovascular diseases (e.g., hypertension and stroke), identified through registered doctor’s prescription; (iii) comprehension of the study’s goal; and (iv) consent to participate in this study.

The sample size was determined by assuming a confidence interval of 95% for z = 1.96 and d = 0.05. The sample portion was assumed to be 0.50 because this value results in the largest possible sample population [15]. As a result, the sample size necessary was 384. However, the sample size was determined to be 883 using design effect 2.3. For this study, we examined the design effect in order to enhance the representative sample and minimize errors. The questionnaire was pre-tested with a small, randomly selected group of participants. Based on their responses, the questionnaire was updated to be clearer and more understandable.

A group of trained healthcare professionals was involved in data collection from the outpatient departments of the designated different tertiary hospitals (both for face-to-face and telephone interviews). Face-to-face data collection was conducted with written informed consent and took roughly 10–15 min to complete. The researchers obtained verbal consent for the telephone survey over the phone, which took approximately 15–20 min to finish. A well-structured, closed-ended self-reported questionnaire was created, and the questionnaire was written, reviewed, and finalized by a public health investigator, a psychiatrist, and an epidemiologist. The questionnaire was initially written in English before being translated into Bangla. We began with a bilingual translator, had it verified by an independent consultant, and then had it back-translated again by a multilingual translator to ensure consistency and to eliminate prejudice.

### 2.3. Metrics

The structured questionnaire utilized in this study included questions about informed consent, sociodemographic data, COVID-19-related characteristics, and perceptions of telemedicine.

#### 2.3.1. Sociodemographic Characteristics

Information on participants’ age, gender, marital status, educational levels, occupation, type of family, monthly income, and place of residence was gathered.

#### 2.3.2. Factors Associated with COVID-19

COVID-19-related factors included the following: Have you lately lost a source of income due to COVID-19 (yes/no)? Have you recently lost a close family member due to COVID-19 (yes/no)?

#### 2.3.3. Perceptions Regarding Telemedicine

The section on perceptions regarding telemedicine included nine questions with yes/no responses. These questions assessed the willingness to receive telemedicine services of those with current chronic diseases.

### 2.4. Ethical Clearance

The Institutional Review Board/Ethical Review Committee (IRB/ERC) of North South University granted ethics approval (2020/OR-NSU/IRB-No.0801). Prior to the interview, study participants provided written/verbal consent. The study was conducted in accordance with the Helsinki Declaration’s commitment to human rights protection. Participants were thoroughly instructed on the process and purpose of the study and were informed that their information would be kept confidential and anonymous. Participants were not compensated for their participation in the research and were free to withdraw at any moment without providing evidence. After reading/hearing the study’s objectives, nature, risks, and benefits, participants agreed to take part by ticking/stating “Yes” (I agree) or “No” (I disagree) on the questionnaire prior to viewing/hearing it. Those who refused to consent were not authorized to view or participate in the survey.

### 2.5. Data Analysis

The Statistical Package for Social Sciences (SPSS), IBM Statistics version 25, was used to examine the data. We conducted descriptive analyses (frequencies and percentages) to characterize the sociodemographic characteristics of 878 participants, risk variables for COVID-19, and their replies to the nine-item questionnaire. The continuous variables such as age and years of education were categorized during the primary analysis. Another continuous variable, monthly income, was first dichotomized and then converted from BDT to USD, based on the conversion rate at the time of the study.

The significance of differences between the sociodemographic variables and WRTCCD was evaluated by the Chi-square (χ2) test and Fisher’s exact test. Odds ratios (ORs) were calculated to assess the association between WRTCCD and the independent variables of interest. ORs also indicated the strength of association; in addition to ORs, their 95% confidence intervals (CIs) were also estimated. A binomial logistic regression model was used to analyze the possible determinants of WRTCCD management. All the independent variables of the study were included in the model. If the *p*-value was less than 0.05, the association between variables was considered statistically significant.

## 3. Results

We collected data from 878 adults who had at least one chronic condition. Table 1 summarizes the participants’ sociodemographic characteristics. The participants’ mean age was 50.1 years (SD: 13.7). Most of them were female (53.0%), aged 35–54 years (46.9%), and married (87.2%). Over 50% of the study population (53.0%) had completed ≥10 years of education and had a monthly income of ≤USD116.58 (51.8%); however, one-third of the participants were joint families (35.4%) and living in rural areas (46.2%) (Table 1).

About half of the participants had lost earnings during COVID-19 (46.6%), and only 14.1% of participants had lost their close relatives due to the COVID-19 pandemic (Table 2).

Most of the participants showed a positive attitude towards telemedicine. Among them, 91% of participants thought that telemedicine could save time, and that people with COVID-19 symptoms or diagnoses should use telemedicine for medical care. In total, 90% of the participants were willing to learn about telemedicine, and 86.9% thought that telemedicine had the potential to play an important role in providing healthcare. In this study, 85.3% of participants thought people with chronic diseases should use telemedicine for their mental healthcare. Additionally, more than 60% of participants thought that standard healthcare could be provided via telephone/computer audio or video conferencing, and 72% were willing to receive telemedical care even after the COVID-19 pandemic. Finally, 78.7% of participants were associated with WRTCCD (Table 3).

Age and years of education were significantly associated with WRTCCD, with 82.8% in the 35–54-year-old age group and 85.3% in the 1–5 years of education group. We found that those in the service/business occupation (82.3%) and those with a monthly income > USD 116.58 (85.1%) were significantly associated with WRTCCD. Additionally, people who did not lose their earnings during the COVID-19 pandemic (85.1%) were also significantly associated with WRTCCD (Table 4).

After the binomial logistic regression, we found that people aged 35–54 years were 51% less likely to be willing to use telemedicine compared to the younger (18–34 years old) group. People with 1–9 years of education and no education were, respectively, 2.03 times and 3.81 times more likely to be willing to receive telemedicine services for a chronic disease that is currently being treated, compared to those who were highly educated (>10 years of education). People earning more than USD 116.58 a month were 49% less willing to use telemedicine to treat a present chronic illness, compared to people earning ≤ USD 116.58 per month. People living in rural areas were 39% less strongly associated with WRTCCD than those who resided in urban areas. Moreover, people who lost their earnings due to the pandemic were 46% less strongly associated with WRTCCD compared to their urban counterparts (*p* < 0.001) (Table 5).

## 4. Discussion

This is the first study conducted in Bangladesh that provides an insight into chronic disease patients’ perceptions of telemedicine during COVID-19. Given the pandemic’s impact on this cohort’s health and safety, this research explores NCD patients’ perceptions toward and willingness to employ modern technologies such as telemedicine to manage their chronic illnesses. The findings of this study indicate that telemedicine is often regarded as a highly satisfactory method of receiving care in the field of chronic disease. The majority of participants regarded telemedicine favorably, emphasizing benefits such as time savings. This approach could be due to the time required to visit a physician at any healthcare facility in this country [16]. Most of the participants also thought that people with a COVID-19 infection or symptoms should use telemedicine rather than a face-to-face visit, most likely because they are afraid of spreading the infection among the high-risk group. Moreover, most of the patients in our study were willing to learn how to use telemedicine to receive medical care.

In our logistic regression analysis, we found that people with no education and less than 10 years of education were significantly more WRTCCD compared to the people with more than 10 years of education. Similar findings were found in the United States, where those with lower levels of education accessed telemedicine at a higher rate as the pandemic progressed [17]. This trend might be due to patients with a higher level of education being more concerned about the privacy issues involved with telemedicine.

Additionally, younger people (18–34 years old) were significantly more likely to be WRTCCD compared to the older group (35–54 years old). Previous research has found that older people are less likely than younger people to use modern gadgets such as computers and the Internet [18,19]. This attitude could be because older people are less confident about using the Internet, particularly an Internet-based telemedicine system that they do not regularly use [20].

Another significant finding of our study is that people with higher incomes were more WRTCCD compared to lower-income people. A similar finding has been observed in previous studies where individuals with a higher income seemed to be more likely to utilize telemedicine services [21,22]. This finding implies that, while telemedicine has both potential for the future and the potential to address healthcare disparities, its technological prerequisites are neither universally accessible nor affordable [23]. This trend was further observed in a US study, where patients who did not choose telemedicine over in-person sessions were much poorer [24]. Additionally, our study found that people who did not lose their earnings due to the COVID-19 pandemic were more WRTCCD compared to people who lost their earnings, a finding that is consistent with previous studies [23,24,25].

Finally, our study found that people living in urban areas were more likely to use telemedicine compared to people living in rural areas. A similar finding was observed during the same timeframe in the US, where those living in rural locations were less willing to use telehealth than those living in metropolitan areas [2]. It is expected that telemedicine healthcare requires possessing a computer, access to the Internet, or a mobile telephone, all of which are more easily and efficiently available in urban areas compared to rural areas.

This study was conducted to ascertain patients’ perceptions of telemedicine and offer policymakers the information necessary to develop an action strategy and direction. However, there are a couple of limitations. Firstly, the preference assessments were conducted in a “yes or no” style, which constrained their interpretation when translated to actual attitudes. Another limitation is that developing a direction of causality under investigation was impossible due to the data-gathering method presenting only a snapshot. However, because our study included a sufficient sample from numerous tertiary hospitals in Dhaka, the likelihood of generalizing the findings to the entire city is increased. As a result, more studies employing alternative approaches and including participants from diverse regional areas may generate different conclusions. As a result, the findings are not confined to a specific intervention, but rather indicate some of the critical components that must be considered when developing and implementing future telehealth efforts.

## 5. Conclusions

The key strength of this study is that it covers a wide range of patient interests in various types of telehealth. In Bangladesh, there is an expansive opportunity for telemedicine to be incorporated into the existing healthcare system if appropriate training and education are provided for healthcare professionals. Before integrating telemedicine services into health systems, it will be essential to improve users’ understanding of how the technology works and how to adapt to it easily. It will also be important to make telemedicine services user-friendly, especially considering socio-cultural and demographic factors (such as local language and rural residents).

## Figures and Tables

**Table 1 ijerph-19-04250-t001:** Sociodemographic information of the study participants (N = 878).

Variables	Number of Study Participants	Percentage (%)
Age (mean ± SD)	50.10 (13.73)	
18–34 years old	113	12.9
35–54 years old	412	46.9
≥55 years old	353	40.2
Gender		
Male	413	47.0
Female	465	53.0
Marital Status		
Unmarried/divorced/widowed	112	12.8
Married	766	87.2
Family type		
Nuclear	567	64.6
Joint	311	35.4
Years of Education		
≥10 years	465	53.0
1–9 years	331	37.7
No education	82	9.3
Monthly Income Range		
≤USD 116.58	455	51.8
>USD 116.58	423	48.2
Occupation		
Homemaker	399	45.4
Service/businessman	345	39.3
Student/retired/other	134	15.3
Residence		
Urban	472	53.8
Rural	406	46.2
Chronic Diseases *		
Diabetes	472	53.8
Hypertension	421	47.9
Cancer	102	11.6
Heart Disease	89	10.1
Respiratory Diseases	71	8.1
Others	48	5.5

* Multiple answers.

**Table 2 ijerph-19-04250-t002:** Impact of COVID-19 on the study population (N = 878).

Variable	Number of Study Participants with Positive Responses	Percentage (%)
Did you lose your earning due to the COVID-19 pandemic?	409	46.6
Did you recently lose any of your close family members due to the COVID-19 pandemic?	124	14.1

**Table 3 ijerph-19-04250-t003:** Participants’ replies to the nine-item questionnaire (%) (N = 878).

Items of Questionnaire	Number of Study Participants with Positive Responses	Percentage (%)
Item (1): Willing to learn about the use of telemedicine for getting healthcare	788	89.7
Item (2): Thinks telemedicine saves time in case of medical visits and follow-up	800	91.1
Item (3): Thinks chronic disease can be managed properly through video conferencing	632	72.0
Item (4): Willing to receive telemedicine service for current chronic disease	691	78.7
Item (5): Thinks people with COVID-19 symptoms or who are COVID-19 positive should use telemedicine for medical care	803	91.5
Item (6): Thinks people with chronic diseases should use telemedicine for their mental healthcare-related advice/counseling	749	85.3
Item (7): Thinks doctors can provide standard healthcare via telephone/computer audio or video conferencing	568	64.7
Item (8): Thinks telemedicine has the possibility to play an important role for providing healthcare to Bangladeshi public	763	86.9
Item (9): Willing to receive healthcare via telemedicine if needed after the COVID-19 pandemic	632	72.0

**Table 4 ijerph-19-04250-t004:** Association between WRTCCD with sociodemographic characteristics (N = 878).

Characteristics		WRTCCD Management		*p* Value
		Yes	No	
Age	18–34 years old	77 (68.1)	36 (31.9)	0.003
	35–54 years old	341 (82.8)	71 (17.2)	
	≥55 years old	273 (77.3)	80 (22.7)	
Gender	Male	328 (79.4)	85 (20.6)	0.680
	Female	363 (78.1)	102 (21.9)	
Years of Education	≥10 years	366 (78.7)	99 (21.3)	0.047
	6–9 years	128 (76.2)	40 (23.8)	
	1–5 years	139 (85.3)	249 (14.7)	
	No education	58 (70.7)	24 (29.3)	
Marital Status	Unmarried	24 (64.9)	13 (35.1)	0.261
	Married	607 (79.2)	178 (20.8)	
	Divorced	3 (75.0)	1 (25.0)	
	Widowed/widower	57 (80.3)	14 (19.7)	
Family type	Nuclear	457 (80.6)	110 (19.4)	0.064
	Joint	234 (75.2)	77 (24.8)	
Monthly income (USD)	≤116.58	331 (72.7)	124 (27.3)	<0.001
	>116.58	360 (85.1)	63 (14.9)	
Occupation	Homemaker	313 (78.4)	86 (21.6)	0.014
	Service/business	284 (82.3)	61 (17.7)	
	Student/retired/other	94. (70.1)	40 (29.9)	
Residence	Urban	360 (76.3)	112 (23.7)	0.069
	Rural	331 (81.5)	75 (18.5)	
Chronic disease	1 or None	468 (80.1)	116 (19.9)	0.162
	>1	223 (75.9)	71 (24.1)	
Lost your earning	Yes	343 (73.1)	126 (26.9)	<0.001
	No	348 (85.1)	61 (14.9)	
Death of close family members	No	593 (78.6)	161 (21.4)	1.000
	Yes	98 (79.0)	26 (21.0)	

**Table 5 ijerph-19-04250-t005:** Binomial logistic regression analysis exploring factors associated with WRTCCD management.

Variables	Adjusted Odds Ratio (95% CI)	*p* Value
Age		
18–34 years old	Reference	
35–54 years old	0.49 (0.29–0.83)	0.008
≥55 years old	1.05 (0.70–1.59)	0.697
Sex		
Male	Reference	
Female	0.76 (0.46–1.26)	0.287
Marital Status		
Unmarried/divorced/widow	Reference	
Married	1.09 (0.64–1.85)	0.745
Years of Education		
≥10 years	Reference	
1–9 years	2.03 (1.12–3.70)	0.020
No Education	1.81 (1.01–3.23)	0.045
Family type		
Nuclear	Reference	
Joint	1.37 (0.95–1.98)	0.089
Monthly income		
≤USD 116.58	Reference	
>USD 116.58	0.51 (0.33–0.78)	0.002
Occupation		
Homemaker	Reference	
Service/business	0.11 (0.62–2.01)	0.724
Student/retired/other	1.29 (0.73–2.27)	0.390
Residence		
Urban	Reference	
Rural	0.61 (0.41–0.90)	0.012
Lost earnings due to pandemic		
No	Reference	
Yes	0.54 (0.37–0.79)	0.001
Lost close family members due to pandemic		
No	Reference	
Yes	1.13 (0.68–1.88)	0.630

## Data Availability

The data presented in this study are available from the corresponding author upon reasonable request. The data are not publicly available because of the need to maintain the participants’ anonymity and data confidentiality.

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
