# Peer review of "Perceptions about Telemedicine among Populations with Chronic Diseases amid COVID-19: Data from a Cross-Sectional Survey"

_ijerph, 2022, doi:10.3390/ijerph19074250_

Round 1

Reviewer 1 Report

This topic is interesting and relevant, however, the M&M section is far too weak. The M&M section is the major role of judging the validity and credibility, while this part is too weak to charge. The face-to-face interviews seem not to essential for this study during pandemic period. The method used, such as snow ball sampling seems to be  a qualitative technique, where there is no reason justified for this study. There are many authors listed but little contribution has been made in this study, which makes this study problematic. References need to be enriched and updated. 

Author Response

Comments and Suggestions for Authors:

Comment: This topic is interesting and relevant, however, the M&M section is far too weak. The M&M section is the major role of judging the validity and credibility, while this part is too weak to charge. The face-to-face interviews seem not to essential for this study during pandemic period. The method used, such as snow ball sampling seems to be a qualitative technique, where there is no reason justified for this study. There are many authors listed but little contribution has been made in this study, which makes this study problematic. References need to be enriched and updated.

Response: Thank you for your critical observations and valuable suggestions. We vigorously revised our manuscript. We did face-to-face interviews with only those participants who paid a visit to those public tertiary hospitals in Bangladesh for their illnesses during the pandemic. Other participants were interviewed over the phone using snowball sampling procedures.

We have revised the references and updated 10 recent and most relevant references.

Reviewer 2 Report

Dear Authors,

Please find my comments concerning your paper:

Abstract

  • Goal of the study is not specified.
  • Resume the Conclusions of the paper
  • Novelty of the paper is not underlined

Introduction

  • An overview on current research on the subject is not properly presented:
    • The state of the art in the use of telemedicine in chronic diseases is described, but the references are rather old;
    • Covid 19 is a recent subject, but there is already a lot of work on this topic;
  • The originality/novelty of the paper is not presented;
  • The content of the paper, by chapters, is not detailed;

Materials and Methods

  • The aim of the interview is not specified;
  • Statistics should be described more intensively, particularly in method part;

Results

  • Findings section only reiterated information presented in tables based on data generated from the closed- ended questions;
  • The results must be used with caution because of several limitations underlined in the paper;

Discussion

  • It is not clear from the discussion section what this paper adds to the existing literature. The findings discussed are not that surprising and largely in line with what has already been well established in numerous other studies;
  • Admittedly, my own knowledge of the literature is quite limited to the EU context, but by the authors’ own admission these observations have also been documented in countries such as developing countries [8], Cambodia, Myanmar, Vietnam [10], and Bangladesh [11];
  • The findings are compared only with US and not with a wider coverage;

References

  • Are quite old, more than half being prior to 2017. Only those related to Covid 19 are recent.

For these reasons, I am not convinced that the paper in its current form is sufficiently novel to merit publication in an international journal such as International Journal of Environmental Research and Public Health (IJERPH). This is not to detract from the importance of the findings at a local level.

That is why I recommend that the paper be reconsidered after a major revision.

Author Response

Comments and Suggestions for Authors:

Comments: Abstract

  • Goal of the study is not specified.
  • Resume the Conclusions of the paper
  • Novelty of the paper is not underlined

Response: The authors have revised the abstract and clearly mentioned the goal of the study and the conclusion part concretely as the reviewer suggested.

Comments: Introduction

  • An overview on current research on the subject is not properly presented:
    • The state of the art in the use of telemedicine in chronic diseases is described, but the references are rather old;
    • Covid 19 is a recent subject, but there is already a lot of work on this topic;
  • The originality/novelty of the paper is not presented;
  • The content of the paper, by chapters, is not detailed;

Response: Thank you for your valuable comments. The authors have revised the paper and updated the references with recent related references. Please see the revised manuscript.

Comments: Materials and Methods

  • The aim of the interview is not specified;
  • Statistics should be described more intensively, particularly in method part;   

Response: The authors have revised and mentioned the aim in the study introduction part and also revised the data analysis part.

Comments: Results

  • Findings section only reiterated information presented in tables based on data generated from the closed- ended questions;
  • The results must be used with caution because of several limitations underlined in the paper;

Response: The authors have revised and edited the Results section.

Comments: Discussion

  • It is not clear from the discussion section what this paper adds to the existing literature. The findings discussed are not that surprising and largely in line with what has already been well established in numerous other studies;
  • Admittedly, my own knowledge of the literature is quite limited to the EU context, but by the authors’ own admission these observations have also been documented in countries such as developing countries [8], Cambodia, Myanmar, Vietnam [10], and Bangladesh [11];
  • The findings are compared only with US and not with a wider coverage;

Response: We have revised the references and added more recent and relevant references which can cover widely.

Comment: References

  • Are quite old, more than half being prior to 2017. Only those related to Covid 19 are recent. For these reasons, I am not convinced that the paper in its current form is sufficiently novel to merit publication in an international journal such as International Journal of Environmental Research and Public Health (IJERPH). This is not to detract from the importance of the findings at a local level.

Response: We have revised and updated recent 10 recent references.

Comment: That is why I recommend that the paper be reconsidered after a major revision.

Response: We have tried a vigorous major revision according to your suggestions. Thank you very much.

Reviewer 3 Report

Interesting work on a current topic of crucial importance to health systems. However, the work presents serious flaws in terms of the interpretation of results that condition the conclusions.

As examples of improvements needed:

Materials & Methods

Line 97

Why construct a questionnaire in English? Please state the rationale for doing it.

Results

Lines 198-205

Seems a little confusing to follow the presentation of results. Please, rewrite it: a suggestion to describe Odds Ratio in the same direction (>0 or <0). Also, a strong suggestion is to check the interpretation of the odds ratio. For example, from table 4 it appears that persons with higher income are favorable to WRTCCD (85.1% vs 72.7%); however, in this paragraph is stated that “People with higher income were 0.51 times less likely…”

Author Response

Comments and Suggestions for Authors:

Comments: Interesting work on a current topic of crucial importance to health systems. However, the work presents serious flaws in terms of the interpretation of results that condition the conclusions.

As examples of improvements needed:

Materials & Methods

Response: The authors have revised the manuscript vigorously including ‘Materials & Methods’  section, and tried to address your suggestions.

Comments: Line 97

Why construct a questionnaire in English? Please state the rationale for doing it.

Response: Kindly note that submitting proposals along with questionnaires in English and native language is needed for the IRB submission.

Comments: Results

Lines 198-205

Seems a little confusing to follow the presentation of results. Please, rewrite it: a suggestion to describe Odds Ratio in the same direction (>0 or <0). Also, a strong suggestion is to check the interpretation of the odds ratio. For example, from table 4 it appears that persons with higher income are favorable to WRTCCD (85.1% vs 72.7%); however, in this paragraph is stated that “People with higher income were 0.51 times less likely…”

Response: In the ‘Results’ section, we have revised and rewritten this said expression. Thank you very much for your helpful recommendations. 

Round 2

Reviewer 1 Report

Dear Authors, 

Thanks for your reply and modification. 

Please highlight your research contribution, significance and strengths.

The simplicity of M&M and result section need to be improved. Please list the data collection procedure to ensure its validity. 

Please also describe your data analysis procedure for its credibility.

As there are many authors and participants offering inputs to this study, therefore we are expecting significant contributions in the tele-medicine field.  Regarding the references, there are some significant references are still missing. 

Combi C, Pozzani G, Pozzi G. Telemedicine for developing countries. Applied clinical informatics. 2016;7(04):1025-50.

Sonia Chien-I. Chen; Chenglian Liu. 2020. "Factors Influencing the Application of Connected Health in Remote Areas, Taiwan: A Qualitative Pilot Study." International Journal of Environmental Research and Public Health 17, no. 4: 1282.

Calton B, Abedini N, Fratkin M. Telemedicine in the time of coronavirus. Journal of Pain and Symptom Management. 2020 Jul 1;60(1):e12-4.

Author Response

Reviewer 1:

Comments and Suggestions for Authors:

Comment: Dear Authors,

Thanks for your reply and modification.

Please highlight your research contribution, significance and strengths.

Response: Thank you for your further valuable suggestions. This time, we again revised our manuscript seriously. We have added and highlighted our research contribution, significance, and strengths in the Abstract, Introduction, and Discussion sections.

Comment: The simplicity of M&M and result section need to be improved.

Response: We have revised the ‘Materials and Methods’ and ‘Results’ sections to explain and improve the research protocol and findings clearly.  

Comment: Please list the data collection procedure to ensure its validity.

Response: The data analysis procedure has been described in ‘Participants and Setting’ sub-section in the Materials and Methods part. Addressing the comment, more details have been added.

Comment: As there are many authors and participants offering inputs to this study, therefore we are expecting significant contributions in the tele-medicine field. Regarding the references, there are some significant references are still missing.

Combi C, Pozzani G, Pozzi G. Telemedicine for developing countries. Applied clinical informatics. 2016;7(04):1025-50.

Sonia Chien-I. Chen; Chenglian Liu. 2020. "Factors Influencing the Application of Connected Health in Remote Areas, Taiwan: A Qualitative Pilot Study." International Journal of Environmental Research and Public Health 17, no. 4: 1282.

Calton B, Abedini N, Fratkin M. Telemedicine in the time of coronavirus. Journal of Pain and Symptom Management. 2020 Jul 1;60(1):e12-4.

Response: Thank you for guiding us to choose the most relevant references. We have revised the references and added more references as per your instruction.

Reviewer 2 Report

I think that the authors resolved most of the observations included in my first revision of their paper.

Please find my comments concerning your revised paper below:

  • The novelty of the paper is not emphasizes in the Abstract or Introduction, but only in lines 278-283 of the Discussion chapter. I think that it would be better for it to be included in the Introduction. This was, it would not be necessary to repeat it in the Discussion chapter.
  • The content of the paper, by chapters, is not detailed in the Introduction.
  • The acronym WRTCCD is defined twice, in lines 44-45 and 144-145. Please only keep the definition from lines 44-45.

So, a minor revision is still required.

Author Response

Reviewer 2:

Comments and Suggestions for Authors:

Comments: I think that the authors resolved most of the observations included in my first revision of their paper.

Please find my comments concerning your revised paper below:

The novelty of the paper is not emphasizes in the Abstract or Introduction, but only in lines 278-283 of the Discussion chapter. I think that it would be better for it to be included in the Introduction. This was, it would not be necessary to repeat it in the Discussion chapter.

Response: Thank you for your helpful comments to improve our paper. This time, we have focused on the novelty and added in the Abstract, Introduction, and Discussion sections without repetition.

Comments: The content of the paper, by chapters, is not detailed in the Introduction.

Response: We have revised the Introduction section.

Comments: The acronym WRTCCD is defined twice, in lines 44-45 and 144-145. Please only keep the definition from lines 44-45.

So, a minor revision is still required.

Response: As you suggested, we have deleted the acronym WRTCCD in lines 144-145. Thank you very much.

Reviewer 3 Report

Failures in the interpretation of the odds ratio continue to be visible in the text (lines 200 to 226), compromising the conclusions. It is unacceptable given the results presented in Table 4 and Table 5 that the authors state in the discussion section (line 261) that "people living in urban areas are more likely to use telemedicine compared to the people living in rural areas". This last example is just one of the misinterpretations of the odds ratio. Perhaps, the authors didn't understand my previous remarks.

Author Response

Reviewer 3:

Comments and Suggestions for Authors:

Comments: Failures in the interpretation of the odds ratio continue to be visible in the text (lines 200 to 226), compromising the conclusions. It is unacceptable given the results presented in Table 4 and Table 5 that the authors state in the discussion section (line 261) that "people living in urban areas are more likely to use telemedicine compared to the people living in rural areas". This last example is just one of the misinterpretations of the odds ratio. Perhaps, the authors didn't understand my previous remarks.

Response: We identified why the results section might be confusing. The result of the binomial regression has been rewritten for clarification. Hopefully, this time the results will reflect the findings of the study. Thank you very much for your important guidance.